# Accurate high throughput alignment via line sweep-based seed processing

Markus Schmidt [1], Klaus Heese [2] & Arne Kutzner [1]

Accurate and fast aligners are required to handle the steadily increasing volume of sequencing data. Here we present an approach allowing performant alignments of short reads (Illumina) as well as long reads (Pacific Bioscience, Ultralong Oxford Nanopore), while achieving high accuracy, based on a universal three-stage scheme. It is also suitable for the discovery of insertions and deletions that originate from structural variants. We comprehensively compare our approach to other state-of-the-art aligners in order to confirm its performance with respect to accuracy and runtime. As part of our algorithmic scheme, we introduce two line sweep-based techniques called "strip of consideration" and "seed harmonization". These techniques represent a replacement for chaining and do not rely on any specially tailored data structures. Additionally, we propose a refined form of seeding on the foundation of the FMD-index.

[1] Department of Information Systems, College of Computer Science, Hanyang University, 222 Wangsimni-ro, Seongdong-gu, Seoul 133-791, Republic of Korea. [2] Graduate School of Biomedical Science and Engineering, Hanyang University, 222 Wangsimni-ro, Seongdong-gu, Seoul 133-791, Republic of Korea. Correspondence and requests for materials should be addressed to A.K. (email: kutzner@hanyang.ac.kr)

Within computational genomics, there are various types of alignment problems that can be classified as shown in Fig. 1. These different types of problems require different types of algorithmic approaches in order to be solved efficiently. Applications that implement such algorithmic approaches are called aligners. The approach presented here is for the alignment of shorter sequences with respect to a single genome. Usually, these shorter sequences are the output of a DNA sequencing system and are called reads. Alignments of such reads with respect to a genome are called sequence-to-genome alignments. In contrast to genome-to-genome alignments[1–3], sequence-to-genome alignments typically do not require the consideration of structural rearrangements. Smith-Waterman's algorithm[4] provides optimal sequence-to-genome alignments on the foundation of dynamic programming. However, due to its high computational costs, this algorithm is not suitable for high-throughput sequence alignment, where a large amount of reads (up to several million reads) has to be processed in a reasonable amount of time. For this purpose, several specially tailored aligners have been proposed in recent years[5–10]. These aligners aim at getting optimal alignments, but do not guarantee actually obtaining them. They can be categorized according to the read sizes that they are intended for. Short read aligners are meant to align reads from 36 nucleotides (nt) up to a few hundred nt. Long read aligners are meant to align reads of a few hundred nt up to several thousand nt or even up to several ten thousand nt. Reads that exceed 40,000 nt are possible already[11]. Instances of short read aligners are SOAP2[12], GEM[10], Bowtie 2[6] and BWA-MEM[5]. For long read aligners this list can be extended by Minimap2[9], BLASR[7], GraphMap[8] and NGMLR[13]. In an initial step, which is called seeding[14], most of these aligners search for short matches (called seeds) between read and genome. These seeds may be non-consecutive matches[15–17]. There are two popular approaches for seeding[18]. It can either be done using some index as e.g. the FMD-index[19] or on the foundation of hash tables. The FMD-index allows the computation of supermaximal exact matches (SMEMs)[19]. Seeding with hash tables can be achieved in a memory efficient fashion by using minimizers[20]. A magnitude of techniques has been devised for seed processing that follows the initial seeding. Notable here are the popular chaining[21–23] approach and the line sweep paradigm. Chaining is used in many state-of-the-art aligners as e.g. BWA-MEM[5], Minimap2[9] and Bowtie 2[6], while the line sweep paradigm is exploited in e.g. Mashmap[24] for the computation of homology maps between genomes.

Sequence-to-genome alignment is next to sequence-to-sequence alignment in Fig. 1. In contrast to sequence-to-genome alignments, the aligned sequences are of similar sizes there. A plain pairwise alignment represents the simplest variant of a sequence-to-sequence alignment. It can be local or global. A global alignment aligns two sequences with respect to their entire length. With local alignments, dissimilar parts at the start and end of both sequences may be omitted. Many sequence-to-genome aligners, e.g. Bowtie 2[6], BWA-MEM[5], Minimap2[9] and BLASR[7], rely on sequence-to-sequence alignments for filling gaps or for alignment extension purposes. As for sequence-to-genome alignments, it is possible to rely on dynamic programming[25] for finding optimal alignments here. But, such optimal alignments are computationally expensive.

Aside of pairwise alignments, there are two additional subcategories within the family of sequence-to-sequence alignments: With multiple sequence alignments more than two sequences are aligned simultaneously[26]. Divergent homology detection tries to identify statistically significant matches to a query sequence in a database of reference sequences[27–31].

Here we introduce an algorithmic scheme for efficiently performing sequence-to-genome alignments for read sizes ranging from 100 nt up to several ten thousand nt. The proposed scheme allows high throughput alignments and shows an excellent overall accuracy compared to other state-of-the-art aligners as reported in the results section. Particularly, for alignments comprising many insertions and deletions our approach shows superiority compared to others. As part of our scheme, we propose a refined form of seeding using the FMD-index. This refined form of seeding relies on a divide-and-conquer technique and computes seeds of higher relevance than SMEMs as mentioned in the discussion. Further, we propose two line-sweep-based techniques called strip of consideration (SoC) and seed harmonization for quickly identifying promising positions on the genome and efficiently computing sets of consistent seeds. In combined form, these two techniques represent a drop-in replacement for chaining. Further, our approach does not rely on specially tailored data structures and it can be described concisely in pseudocode. Using generated reads, we comprehensively compare several state-of-the-art aligners with each other and with our approach. An open source application called MA (Modular Aligner) implements our approach. MA has a modular design and is publically available via GitHub at https://github.com/ITBE-Lab/MA.

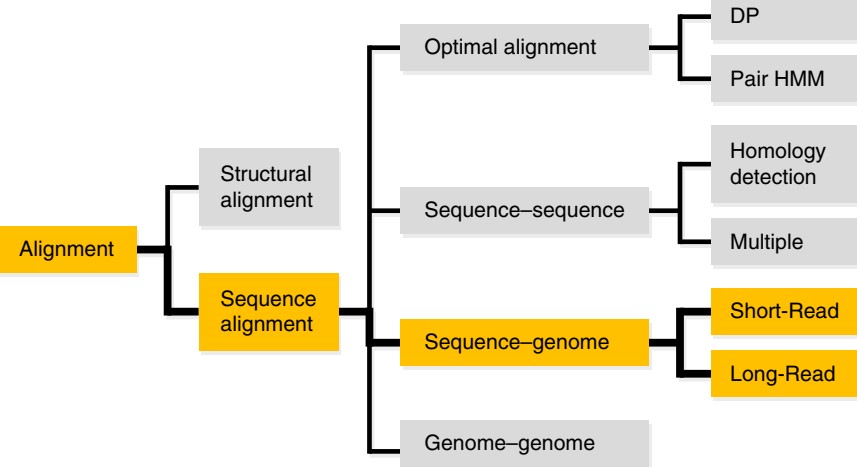

**Fig. 1** Classification of alignment problems. The hierarchy classifies various forms of alignment problems. Our proposed approach belongs to the yellow marked subgraph of the hierarchy

## Results

**Approach outline.** Our approach performs alignments by completing the following three stages: (1) Seeding, (2) seed processing and (3) dynamic programming (DP). Seeding finds perfectly matching substrings between a reference and a query. We do this using the FMD-index[19], which represents a full-text substring index. In stage 2, the seeds obtained in stage 1 are filtered and harmonized, which results in a consistent and spatially local set of seeds. Remaining gaps between seeds are filled using DP[25] in stage 3. Apart from the FMD-index, our approach does not require any specially tailored data structures.

Stage 1: Seeding on the foundation of an FMD-index requires an extension process[19,32]. Our seeding follows a divide and conquer approach that starts at the query center. By applying maximal extensions, a query section (initially the entire query) is decomposed into three intervals, where the central interval is defined by two or more maximally spanning seeds. These maximally spanning seeds are computed in a two-step process. First, we maximally extend forwards and then we maximally extend backwards. By doing so, we obtain one or more seeds that all cover the same query interval. Then, we apply the opposite order of maximal extensions for getting one or more additional seeds. The intervals remaining on the left and right are processed recursively. This approach creates less irrelevant seeds than other seeding approaches[6,7,19] as reported in the discussion.

Stage 2: Informally, two seeds are consistent, if they appear in the same order on reference and query. Further, two seeds are spatially local, if the gap between them is small enough so that an alignment comprising both seeds can have a positive score. In stage 2, sets of consistent spatially local seeds are computed in order to get alignments with optimal scores. We call this stage seed processing. Seed processing can be done by chaining[21–23,33] but our approach does not rely on this technique. Instead, our approach does seed processing in two steps called strip of consideration (SoC) and seed harmonization. The SoC identifies best-scored sets of spatially local seeds. Although not algorithmically realized this way, the computation of SoCs can be imagined as follows: By sliding a fixed size window along the reference, we search for the best-scored window positions, where the score of a position is the accumulative length of all seeds within the window on that position. The overall time complexity of the SoC computational is limited by the complexity of an initial seed sorting. If the index used for seed generation is able to deliver the seeds in correct order, then the SoC can be computed in a single pass in linear time.

Seed harmonization computes a consistent subset of a SoC-seed set. As opposed to chaining, seed harmonization represents a purging technique. In order to be consistent, two seeds $s$ and $s'$ must fulfill one of the following two properties exclusively: Either, the query and reference start positions of $s$ are smaller than the corresponding start positions of $s'$, or, the query and reference end positions of $s$ are larger than the corresponding end positions of $s'$. In the case of a pair of inconsistent seeds, we purge one of them, where we rely on the following decision principle: Using a heuristic approach, we define a line, called $\delta$-guideline that roughly approximates the expected shape of the alignment. The seed showing the larger distance to this line is purged, while the closer seed is kept.

Theoretically, our purging approach could have a squared complexity behavior. However, it requires highly specialized examples to obtain such a worst-case behavior. Practically, the runtimes are driven by an initial seed sorting that is required as part of the harmonization.

Stage 3: For computing optimal alignments, there are several DP algorithms[4,25,34]. They are all controlled by some scoring scheme. Like other long read aligners[5–7], we make use of banded DP[35,36]. We primarily use it for filling small gaps occurring between the seeds of a harmonized SoC-seed set. Due to this limited use of DP, its overall computational penalty is proportional to the dissimilarity of query and reference. So, in the case of high seed coverage DP costs become low.

**Aligner comparison based on generated reads.** In the following, we compare MA with several other aligners for short and long reads, where we analyze Minimap2[9], NGMLR[13], GraphMap[8] and Blasr[7] for long reads as well as BWA-MEM[5] and Bowtie 2[6] for short reads. The lack of ground truth with real-world data prevents the direct inference of an alignment's accuracy. Hence, our evaluation relies on simulated reads. For creating Illumina reads, we use DWGSIM[37]. Pacific Biosciences (PacBio) reads and Ultralong Oxford Nanopore (UON) reads are generated from sampled error profiles. These error profiles are computed by evaluating alignments obtained via Minimap2 for the datasets PacBio-MtSinai-NIST and Ultralong Oxford Nanopore (UON) of HG002 (AJ Son)[38]. Our approach resembles the technique used by SURVIVOR[13] and is comprehensively described in Supplementary Note 1. Opposed to SURVIVOR, our approach incorporates distributions for the observed lengths of insertions and deletions. This boosts the resemblance among generated reads and real-world sequencer reads. The bar diagrams at the bottom of Fig. 2 depict the distributions of reads with respect to the mean error rates.

We now discuss the behavior of all inspected aligners. For long reads (PacBio and UON), all aligners deliver a similar high accuracy within the left third of the diagrams. Then, the accuracy deteriorates with different speeds, where MA is only surpassed by GraphMap. Further, there are significant differences with respect to the runtime, where Minimap2 and MA belong to a more performant group than GraphMap, Blasr and NGMLR.

The read distributions with PacBio and UON (as shown by the bar diagrams) indicate the existence of notable amounts of reads within the range of two times up to three times of the mean error. Therefore, the different accuracy levels of the inspected aligners within this range are relevant to their assessment. For all generated reads, we assume that the location of the genome section used as template for the read's generation is the read's only correct position. If a template is chosen from an ambiguous section of the reference genome, an aligner can deliver an alignment for an alternative position of this ambiguous section. Such different positions are assessed as wrong positions, because they are unknown during read generation. This explains the inability of all aligners to reach 100% accuracy for perfect reads for all analyzed datasets.

An evaluation for short reads (paired Illumina reads) is given in Fig. 3. Compared to long reads (PacBio and UON), Illumina reads are less prone to sequencing errors, but they are more likely to deliver ambiguous alignments. Therefore, the mapping quality represents crucial information for this kind of reads. Figure 3 is motivated by similar figures of Li[5] and visualizes the behavior of aligners depending on the mapping quality. The figure shows that MA's mapping behavior is superior to the behavior of Bowtie 2 but inferior to that of BWA-MEM. However, MA is roughly twice as fast as BWA-MEM and three times faster than Bowtie 2. Additional information can be found in Supplementary Note 8. Among all aligners, MA is the only one that suits short as well as long read alignments. As opposed to k-mer based approaches, MA does not require adapted indices for different types of sequencing techniques or genomes (Supplementary Fig. 7). This proves the universal nature of the proposed algorithmic schemes.

Table 1 reports several additional benchmarks for all analyzed aligners. The startup time reflects the initial timespan required by

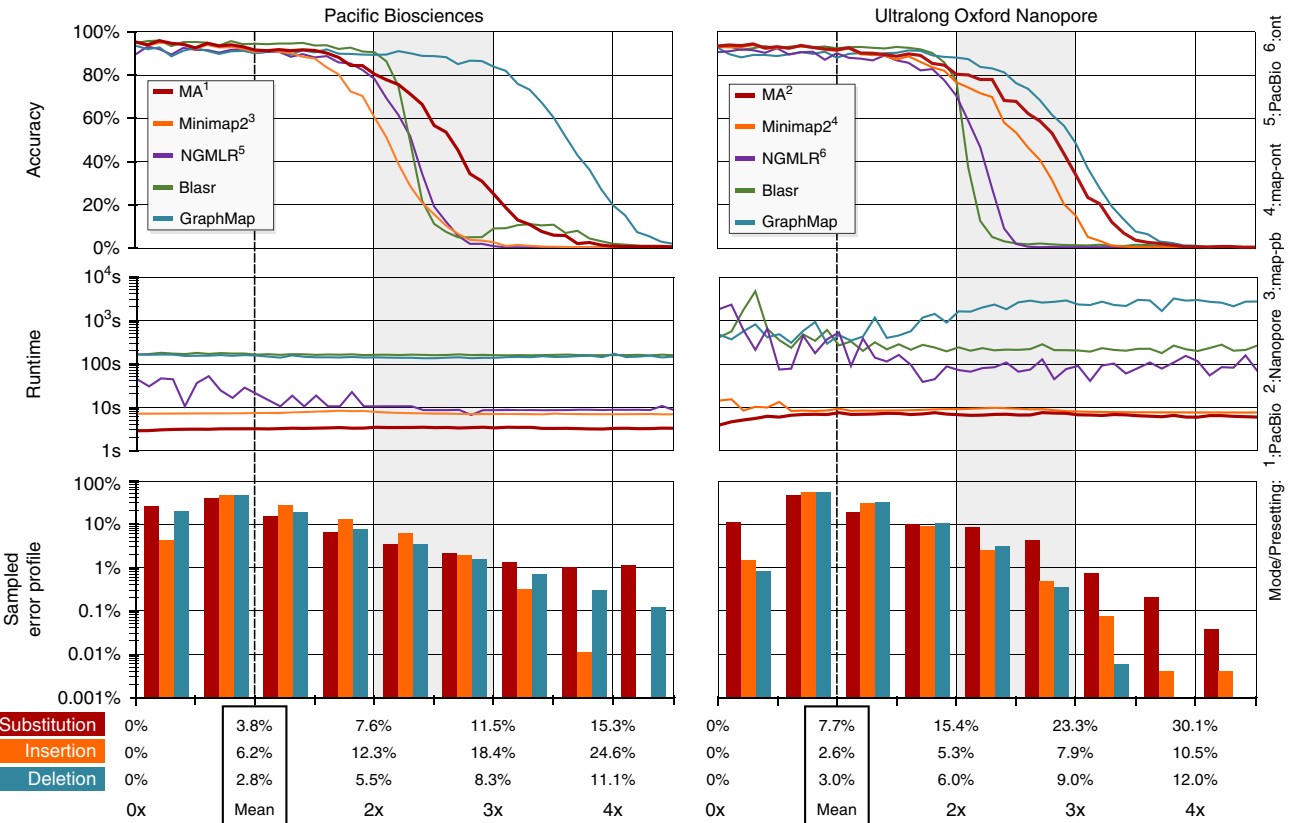

**Fig. 2** Analysis of multiple aligners for various sequencing techniques. The top part of each diagram shows the computed accuracy behavior for various aligners (Minimap2, NGMLR, Blasr, GraphMap, MA). The center part displays the observed total runtimes, where each measurement is for $10^3$ reads. The accuracy is measured in the range from 0 to 4 times of the sampled mean error. For example, the 60% accuracy of Minimap2 with PacBio is measured for reads of 7.6% substitution rate, 12.3% insertion rate and 5.5% deletion rate (twice the mean error). The mean error for the different sequencing techniques is shown in the boxes labeled mean. The bottom part of the diagram depicts the distributions of substitutions, insertions and deletions for various error rates. Each bar spans a range of 0.5 with respect to a scalar extension/shrinking of the three individual components of the mean error. For example, the first red bar indicates that roughly 25% of all PacBio reads have a substitution rate from 0% (error free) up to 1.9% ($=0.5 \cdot 3.8\%$). For insertions (deletions), we count a sequence of consecutively inserted (deleted) nucleotides as one single insertion (deletion). As data source we rely on the GiaB data for HG002[38]. Further details can be found in Supplementary Note 1 and Supplementary Note 8

the aligner before it starts the actual alignment operation. The index size denotes the amount of disk space observed for storing an index of the human genome (GRCh38.p12). The peak memory consumption (Peak mem) is measured using the tool Valgrind[39]. MA can perform long read alignments while still having the footprint of a short read aligner.

**Discovery of indels caused by structural variants**. Now we additionally analyze MA with respect to the mapping of two types of structural variants: insertions and deletions. For this purpose, we generate reads on the foundation of the sampled UON mean error and inject exactly one SV into each generated read. According to an aligner's CIGAR for a generated read, we distinguish among six categories of SV detection accuracy: Precise: The aligner reports the accurate SV position. Split: The SV is reported accurately using two alignments. Indicated: The positon of the SV is misplaced more than 10nt. Forced: The SV is delivered in fragmented form. Trimmed: Reads are clipped near the SV. Unaligned: There is no alignment close to the SV position. A detailed explanation of these categories is given in Supplementary Note 2. Figure 4 shows the outcome of our comparison in form of bar diagrams. (The respective diagrams for PacBio reads are in Supplementary Fig. 6). The diagrams are motivated by the corresponding diagrams of Sedlazeck et al.[13], where our accuracy categorization is similar,

but our measurement approach is different (this strongly effects the category "unaligned"). Despite these differences, both sets of diagrams confirm a low level of precise SV (indel) detection for GraphMap and Blasr. Within the category "precise" of Sedlazeck et al.[13], we distinguish a subcategory "split", where the aligner finds accurate end-points of the SV but reports the SV as chimeric alignment (one primary alignment, one supplementary alignment). The justification for this additional category is the indirect indication of the SV in contrast to the direct indication with "precise". MA recognizes SV via the SoC. Therefore, it produces very few "split" alignments in contrast to Minimap2 and NGMLR. This behavior is advantageous in ambiguous regions as indicated in the discussion. All aligners show a similar decreasing level in SV detection for increasing SV sizes. This is to be expected because with increasing SV size an increasing portion of the read is not related to the reference genome anymore.

**Discussion**
One highlight of our work is the proposed technique for computing sets of consistent spatially local seeds. We compute these sets using the SoC followed by seed harmonization. The efficiency of this backend allows performant alignments of long reads using SMEMs[19] as shown by the runtimes for UON reads in Fig. 2. Using our maximally spanning seeds, MA shows even better performance as indicated by the runtimes for PacBio in Fig. 2.

Aside from this strong efficiency, our backend has several significant theoretical advantages compared to other backend approaches:

Some aligners, e.g., GraphMap, rely on a bucketing technique instead of the SoC. Bucketing techniques have several disadvantages compared to our approach: Bucket-boundary crossing alignments require special consideration. If the size of all buckets is fix, the number of buckets is proportional to the genome size. Therefore, the costs (runtime and memory) for the discovery of the best buckets become proportional to the genome size.

Another replacement for the SoC is chaining, which has received a lot of attention in research already[7,21–23]. In the following, we will compare our technique to chaining.

All chaining techniques share the property that they sequentially grow chains of seeds. This can be done without sophisticated data structures[5,9], but for the price of a squared worst-case complexity. Alternatively, it can be done on the foundation of special tailored data structures in time $O(n \log n \log \log n)$ (with gap-costs[21]) or $O(n \log n)$ (without gap-costs[23]). However, the

latter solutions rely on complex data structures and, to our knowledge, they have not been integrated into any aligner so far. Our technique does not require sophisticated data structures and is, due to its simplicity, highly performant. The time complexity of our SoC computation is driven by the initial sorting of all seeds. For specially constructed examples, the seed harmonization can show a squared worst-case complexity. On average, however, the seed harmonization can be expected to be completed in time $O(n \log n)$, where $n$ is the number of seeds.

The chaining approaches with gap costs[7,21,22] cannot cope well with overlapping seeds, because they cannot concatenate them. The following example shall explain the severity of this limitation: Let the string $AATGG$ be the query and let the string $AATCTGG$ be the reference. There are two non-enclosed (maximally extended) seeds (>1nt) on the reference $\overline{AATC}\underline{TGG}$, which are overlapping on the query $\overline{AA}\underline{TGG}$. (The over- and underlines indicate the locations of both seeds on query and reference). This overlap is triggered by the occurrence of a $T$ on either side of the $C$. Such overlaps occur with a probability of 7/16 for seeds that are separated by indels (1/4 on both sides of the indel; assuming random strings). Hence, such cases should be quite common on nucleotide sequences. Chaining has to drop one of these seeds, because a concatenation is impossible. With our approach, the seed harmonization keeps both seeds. The overlap among them does not cause any trouble.

In the case of pairs of contradicting seeds, one seed has to be chosen in favor of the other seed or both have to be purged. Chaining has to perform the required decision on the foundation of the local context. Several strategies have been proposed for this purpose (e.g. gap-costs[21] or accumulative seed lengths[23]). Figure 5 shows an example, where these strategies run into an inconvenient ambiguity. They cannot distinguish between both paths due to equal gap costs and seed lengths. Our predicted alignment approximation (introduced as $\delta$-guideline) delivers global information about the expected overall alignment path. In our approach, we pick the seed closer to the $\delta$-guideline. This should be advantageous, because an alignment via $s$ allows more substitutions of gaps by matches or mismatches than an alignment via $s'$.

Alignments in repetitive regions of a genome demand for long seeds because short seeds can show unmanageable amounts of repetitive occurrences in these regions (Supplementary Fig. 7). In contrast, noisy reads demand for short seeds because exact matches of an inappropriately chosen size might not lead to sufficiently many seeds, which are required to guarantee the discovery of the correct position. Therefore, fixed-length seeds require sequencer specific fine-tuning for meeting both requirements simultaneously (k-mer sizes must be adapted to sequencing techniques). Variable-length seeds, as e.g. SMEMs[19], do not require such fine-tuning. Additionally, variable-length seeds become longer and decrease in quantity with increasing read quality, while fixed-size seeds are expected to increase in quantity. (With fixed-size seeds, the information about a long perfect match has to be decomposed into many small pieces.)

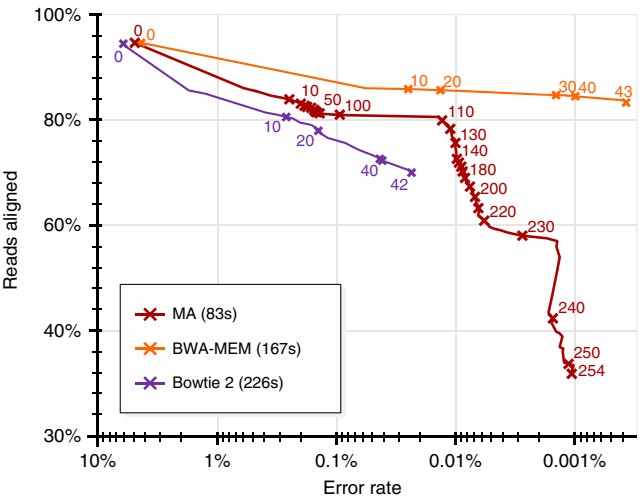

**Fig. 3** Accuracy analysis for paired-end Illumina reads. The diagram shows a comparison of the three aligners BAM-MEM[5], Bowtie 2[6] and MA using 250nt paired-end reads generated by DWGSIM[37]. Each point corresponds to its annotated mapping quality threshold $n$ and reports accumulatively about all alignments having a mapping quality $\geq n$. The y-axis informs about the percentage of aligned reads for a specific mapping quality threshold and the x-axis indicates the relative amount of wrong alignments among those reads. For example, the point annotated with 110 on the red curve expresses the following: MA aligns 80% of all simulated reads with a mapping quality of $\geq 110$ and roughly 0.01% of the corresponding alignments are wrong. By default, 5% of the reads generated by DWGSIM are completely random. For those reads, aligners should not deliver any alignment at all, which explains the lack of 100% alignment rate for all aligners. All runtimes are reported right to the aligner names within the legend. Each curve delivers a statement about $10^6$ read pairs. Further details are given in Supplementary Note 8 and Supplementary Fig. 4

**Table 1 Additional aligner benchmarks**

|  | MA | Minimap2 | NGMLR | Blasr | GraphMap | BWA-MEM | Bowtie 2 |
|---|---|---|---|---|---|---|---|
| Startup time | 2.8 s | 7.1 s | 4.6 s | 143 s | 46.0 s | 2.6 s | 1.9 s |
| Index size | 5.7 Gb | 7.1 Gb | 5.3 Gb | 13 Gb | 41 Gb | 5.7 Gb | 4.4 Gb |
| Peak mem. | 6.0 Gb | 9.2 Gb | 6.4 Gb | 16.4 Gb | 46.2 Gb | 6.1 Gb | 6.1 Gb |

The benchmarks reported in column 1 to 5 are measured using simulated PacBio reads, while column 6 and 7 comprise benchmarks received via Illumina reads generated by DWGSIM. Supplementary Note 8 describes the environment used for this analysis

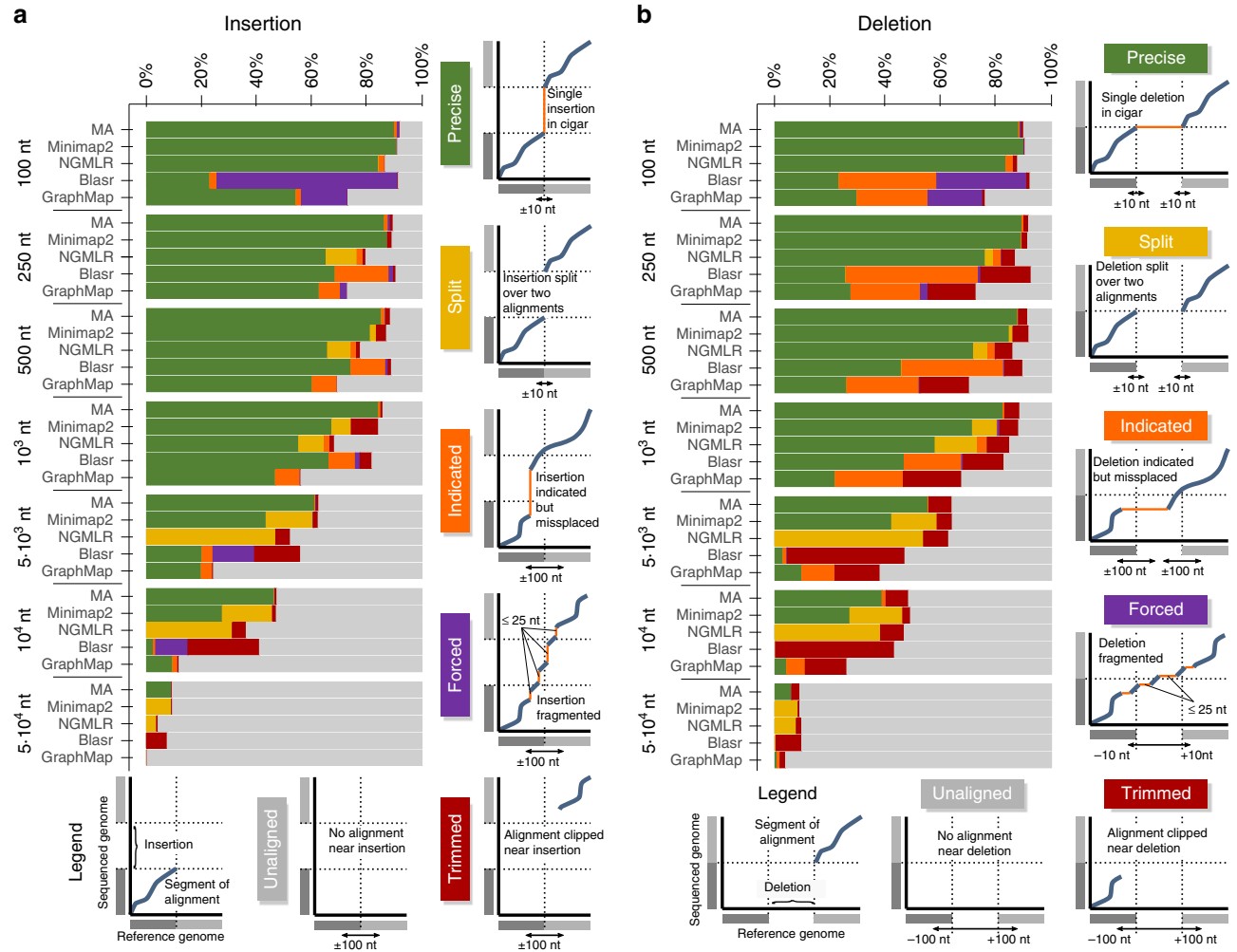

**Fig. 4** SV mapping. Several aligners (MA, Minimap2, NGMLR, Blasr and GraphMap) are compared with respect to their abilities to map insertions (**a**) and deletions (**b**) originating from structural variants (SV). The numbers to the left of the aligner names indicate the sizes of injected SVs. Textual definitions of the different categories precise, split, indicated, trimmed, forced and unaligned are given in Supplementary Note 2. Each bar reports about the observed distribution for 1000 reads that are generated using the sampled mean error of the UON dataset

Maximally spanning seeds constitute a subset of SMEMs[19]. Although being a subset, maximally spanning seeds can deliver similar accuracy to SMEMs thanks to their high relevance (Supplementary Fig. 8). This relevance aspect is now inspected more in detail. A seed $s$ is called relevant with respect to a query $Q$ if and only if $s$ overlaps with the reference interval of $Q$ (i.e., the seed leads to the correct position on the reference). Let $S$ be a set of seeds and let $S' \subseteq S$ be the set of relevant seeds in $S$. We define the relevance rate of $S$ as the ratio $|S'|/|S|$. For simulated UON reads (mean error sampled from HG002 (AJ Son) by Zook et al.[38]), maximally spanning seeds deliver a relevance rate of 4.4% (i.e., 4.4% of all seeds lead to the correct position on the genome), while for SMEMs the rate is 1.5% merely. With simulated PacBio reads, the relevance rates are 5.0% (maximally spanning) and 1.7% (SMEMs).

We inspect MA with respect to insertions and deletions that represent SVs. Apart from indels, there are other forms of structural variants as e.g. duplications, translocations and inversions. Currently, MA does not distinguish between insertions and duplications; it always reports insertions. Translocations can be discovered via SoCs, if the translocated segment forms a separated SoC of sufficient score. Such translocations are reported via supplementary alignments. In the context of the SoC, sufficiently

sized inversions resemble translocations between forward and reverse strand. Therefore, MA can report sufficiently sized inversions via supplementary alignments like translocations. Our SV analysis for insertions and deletions comprises a category "split". "Split" alignments report the SV accurately but indicate that an aligner discovers both ends separately. However, if one of both ends originates from an ambiguous region, the ambiguity can prevent the SV discovery. SoC based seed processing avoids such splitting and so it helps with the precise discovery of SVs in such cases.

Overall, the proposed algorithmic approach allows a modular application architecture as shown by our implementation (see Code availability section). MA's codebase consists of individual modules, realized in C++, which are coupled using a dependency graph for getting a functional aligner. The module coupling can be achieved via Python or C++ itself. MA comprises a graphical user interface application as well as a command line application that are both realized via a C++ based module coupling. The graph-based design allows modules to be interchanged seamlessly, which boosts flexibility and supports experimental work with alternative aligner design. Therefore, MA can be perceived as a general framework for aligner design and construction.

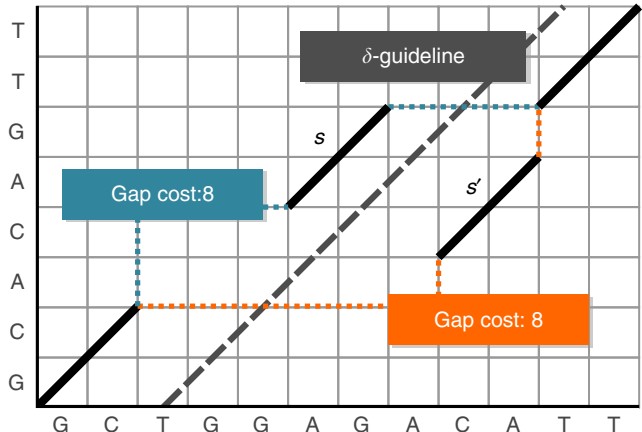

**Fig. 5** Contradicting seeds. A scenario with two contradicting seeds $s$ and $s'$. While extending the chain, chaining[11,27–29] is forced to choose one of both seeds. Abouelhoda et al.[21] use gap cost estimations, while Shibuya et al.[23] use accumulative seed lengths for making this decision. In contrast to our approach, however, both techniques cannot guarantee that the selected seed belongs to the best alignment

## Methods

**Basic notions and planar alignment representation**. We consider alignments as a path that follows a sequence of points $[(r_n, q_n) \mid 0 \le n \le |A|; r_n, q_n \in \mathbb{N}_0]$ in a two-dimensional plane, where $|A|$ shall denote the alignment length. With this plane, the reference $R$ is on the x-axis and the query $Q$ is on the y-axis, with $|Q| < |R|$. Please note: The sequences $r_0, r_1, ..., r_{|A|}$ and $q_0, q_1, ..., q_{|A|}$ are both monotonically increasing. Two consecutive points $(r_n, q_n)$ and $(r_{n+1}, q_{n+1})$ are always associated in one of the following four ways:

- Deletion: $r_{n+1} = r_n + 1$, $q_{n+1} = q_n$; visually this is equal to a horizontal path extension.
- Insertion: $r_{n+1} = r_n$, $q_{n+1} = q_n + 1$; visually this is equal to a vertical path extension.
- Match or Mismatch: $r_{n+1} = r_n + 1$, $q_{n+1} = q_n + 1$; visually both appear as diagonal path extensions.

For example, in Fig. 5 a complete alignment via the orange path follows the sequence: $[(0, 0), (1, 1), (2, 2), (3, 2), (4, 2), (5, 2), (6, 2), (7, 2), (8, 2), (8, 3), (9, 4), (10, 5), (11, 6), (11, 7), (12, 8), (13, 9)]$

The alignment process is decomposed into three separated stages: seeding, seed processing and dynamic programming (DP). In the following three sections, we will explain our approaches for all three stages in detail.

**Seeding**. Seeding computes a set of seeds, where a seed is a perfectly matching section between query and reference. Each seed is represented as a triple $(q, r, l)$, where $q$ and $r$ are the starting positions on query and reference, respectively $(q, r \in \mathbb{N}_0)$. $l$ denotes the length of the seed $(l \in \mathbb{N})$. Hence, a seed describes the equivalence of the interval $[q, q + l]$ on the query with the interval $[r, r + l]$ on the reference. We do not consider spaced seeds or gaped seeds[6,12,15–17].

In order to refrain from scanning the reference, our approach relies on a full-text substring index[40–42]. We denote such an index as $I_R$ and expect it to implement three functions $I_R^*(c)$, $I_R^\leftarrow\left(c, O_{R,s}\right)$ and $I_R^\rightarrow\left(c, O_{R,s}\right)$. We call $I_R^\leftarrow$ and $I_R^\rightarrow$ backward extension and forward extension, respectively. Given a single character $c$ (typically a nucleotide symbol), $I_R^*(c)$ shall return the set $O_{R,c}$ containing the positions of all occurrences of $c$ in $R$. The position of a string is the index of its first character. Given a set $O_{R,s}$ of all positions of the string $s$ in $R$ and a single character $c$, $I_R^\leftarrow\left(c, O_{R,s}\right)$ shall return $O_{R,c\oplus s}$, where $c \oplus s$ denotes the string concatenation of $c$ and $s$. Similarly, $I_R^\rightarrow\left(c, O_{R,s}\right)$ shall return $O_{R,s\oplus c}$. We assume that all three operations are performed in time $O(1)$. All $O_{R,s}$ shall be represented implicitly (e.g. as suffix array intervals[40]) so that runtime does not depend on the size $|O_{R,s}|$. Further, the size $|O_{R,s}|$ must be obtainable in constant time. Our approach is built on the foundation of FMD-indices[19], but it is possible to use other index-models (as e.g., Bi-directional BWT[43] or enhanced suffix arrays[32]) as well. Our code for the FMD-index as well as the pack management is imported from the code of BWA-MEM[5] and adapted to our needs. FMD-indices are based on FM-indices, which form a space-efficient family of indices[40]. FMD-indices are optimized for searching on genome data, where the forward strand and its reverse complement can be expressed in terms of each other. FMD-indices allow extending on both strands simultaneously[19]. Further, because any sequence and its reverse

complement have suffix arrays of equal size, FMD-indices support forward extensions by interval decomposition.

Because our seeding technique relies on an implicit seed representation, it cannot deliberately discover seeds with specific positional properties on the reference. For this reason, the following three properties (1) fully covering, (2) non-enclosed and (3) maximally spanning are defined on the foundation of query positions.

(1) A set of seeds $S$ is fully covering if and only if there is a seed $(q, r, l) \in S$ for each position $i$ on the query $(0 \le i < |Q|)$, so that $q \le i < q + l$. If the reference alphabet $\Sigma_R$ is a proper subset of the query alphabet $\Sigma_Q$, it is impossible to compute a fully covering set of seeds. So, we assume $\sum_Q \subseteq \sum_R$.

(2) A seed $(q, r, l)$ is non-enclosed with respect to a set of seeds $S$ if and only if there is no other seed $(q', r', l') \in S$ with $q' \le q < q + l \le q' + l'$.

(3) We now give two definitions that formalize the outmost query positions reached by a set of seeds $S$: $S.begin := \min\{q \mid (q, r, l) \in S\}$ and $S.end := \max\{q + l \mid (q, r, l) \in S\}$. A set of seeds $S$ is maximally spanning with respect to a query position $i$, if it fulfills the following property: For all $(q, r, l)$ in $S$ we have $q \le i < q + l$ and it is not possible to construct a seed $(q', r', l')$ with $q' \le i < q' + l'$, so that $q' < S.begin$ or $q' + l' > S.end$.

We now propose an algorithm that, for a given query position $i$, computes a set of maximally spanning non-enclosed seeds, where each seed covers $i$. Based on the resulting seeds, we then compute a fully covering set of seeds.

*Maximally spanning seeds.* We call a repeated extension by $I_R^\leftarrow (I_R^\rightarrow)$, starting from a query position $i$ and continuing while $I_R^\leftarrow (I_R^\rightarrow)$ yields a non-empty set of matches, maximal backwards (forwards) extension. We now propose a scheme that efficiently computes a subset of all non-enclosed seeds spanning over a given query position $i$ $(0 \le i < |Q|)$. Starting from position $i$, we first maximally extend backwards and obtain a set of seeds $S$. Then, we extend $S$ maximally forwards. $S$ can comprise one or multiple seeds, where all seeds in $S$ have identical query intervals. By starting from position $i$ again, but using the opposite order of extensions (first maximally forwards and then maximally backwards), we get a second set of seeds $S'$. We return the union $S \cup S'$ as outcome of the extension process. The pseudocode for this algorithmic approach is given by the procedure MAXIMALLY—SPANNING $(I_R, Q, i)$ in Supplementary Note 3. Trivially, MAXIMALLY—SPANNING always delivers a non-empty set of seeds. We now prove that the above extension process always finds maximally spanning seeds with respect to a given start position $i$.

Corollary 1:
Let $S'$ be the set of all seeds $(q', r', l')$ with respect to a query $Q$ and a reference $R$ that span over a given query position $i$ $(0 \le q' \le i < q' + l' \le |Q|)$. Let $S$ be the set of seeds computed by MAXIMALLY—SPANNING$(I_R, Q, i)$. There is no seed $(q', r', l') \in S'$, so that $q' < S.begin$ or $q' + l' > S.end$.

*Proof:* Assume, there is a seed $(q', r', l') \in S'$, so that $q' < S.begin$. This poses a contradiction to the definition of the maximal backwards extension that continues while there is a non-empty set of matches returned by the extension. Likewise, there cannot be a seed with $q' + l' > S.end$. ∎

Similarly to our approach, BWA-MEM uses a seeding algorithm that initially finds non-enclosed seeds covering a given query position[19]. BWA-MEM then uses this algorithm in order to compute a fully covering set of seeds. These seeds are called supermaximal exact matches (SMEMs), where a seed is supermaximal extended if and only if it is non-enclosed. Our maximally spanning seeds represent a subset of SMEMs. We now explain which unrelated (and therefore irrelevant, because they are not part of an optimal alignment) seeds are non-enclosed but not maximally spanning. These unrelated seeds can be recognized by their query location as follows: They are completely covered by two or more longer non-enclosed seeds. Supplementary Fig. 8 graphically illustrates such situations. In an average UON and PacBio alignment, 56.8% and 57.9% of all non-enclosed seeds (SMEMs) are irrelevant and completely covered by longer non-enclosed seeds, respectively. Merely 4.4% (UON) and 4.9% (PacBio) of these irrelevant seeds are comprised within maximally spanning seeds.

*Binary seeding.* Binary seeding computes a fully covering set of seeds. In each step, binary seeding decomposes a given query interval $i_Q = [b, e]$ into three subintervals $i_L = [b, p_1]$, $i_C = [p_1, p_2]$ and $i_R = [p_2, e]$. The initial interval covers the entire query. $i_C$ comprises the query area covered by the seed-set that originates from position $\frac{b+e}{2}$ and is computed by a call of the procedure MAXIMALLY—SPANNING. Binary seeding continues recursively on non-empty $i_L$ and $i_R$. The pseudocode for binary seeding can be found in Supplementary Note 4.

Corollary 2:
Binary seeding terminates for any given query $Q$ and reference index $I_R$.

*Proof:* Let $m = \frac{b+e}{2}$ be the middle of the query interval $i_Q$. Let $S$ be the set of seeds received by the call MAXIMALLY—SPANNING$(I_R, Q, m)$. All seeds in $S$ overlap $m$ and are of size 1 at least. Thus, we have $S.begin \le m$ and $S.end > m$. We decompose $i_Q$ into $i_L = [b, p_1]$, $i_C = [p_1, p_2]$ and $i_R = [p_2, e]$, where $p_1 = S.begin$ and $p_2 = S.end$. Since $p_2 - p_1 > 0$, we get $(p_1 - b) + (e - p_2) < e - b$. Hence, the size of $i_L$ and $i_R$ is smaller than the size of $i_Q$. Calls for size zero return immediately. ∎

**Seed processing**. Seed processing performs the collection of a set of consistent seeds with the intention of acquiring an optimal alignment. It comprises two major steps: Strip of consideration (SoC) and seed harmonization. A SoC comprises a set

of seeds that cover a candidate region for an optimal alignment. Seed harmonization erases inconsistent seeds within a SoC.

*Strip of consideration (SoC).* Informally, two seed that can participate in an alignment with a positive score are called spatially local. In order to define spatial locality formally, we require the SoC width $\lambda$, which is defined as follows:

Let $s_M$ be the score for a match and let $p_O$, $p_E$ be the penalties for opening and extending a gap, respectively. Further, let $|Q|$ denote the length of a given query $Q$. The width of a SoC, denoted by $\lambda$, is computed as follows:

$$\lambda := \frac{s_M|Q| - p_O}{p_E}. \tag{1}$$

The derivation of $\lambda$ can be found in Supplementary Note 5. In order to get an optimal value for $\lambda$ with respect to DP, the parameters $s_M$, $p_O$ and $p_E$ should be equal to the respective parameters in DP.

Two seeds $(q, r, l)$ and $(q', r', l')$ are considered spatially local if and only if $|\delta - \delta'| \leq \lambda$, where $\delta := r - q$ and $\delta' := r' - q'$ (i.e. the distance between two $\delta$-values must be smaller than $\lambda$). Measuring the distances of seeds using their $\delta$-values is done in other aligners (as e.g. GraphMap[8]) as well.

We now explain our algorithmic approach for computing SoCs, which is related to techniques used in the context of convolutional filters in image processing. We assume that all seeds are stored as triples in an array $A_S$. Initially, $A_S$ is ascendingly sorted according to the $\delta$-values of all seeds in $A_S$. Using a left to right scan, where we adjust a running score, we collect all SoCs that do not overlap with another higher scored SoC. This scan is implemented by using two indices with respect to $A_S$. One index indicates the beginning of the SoC and a second index refers to the end of the SoC. The indices are incremented, so that the referred seeds of both indices keep a distance $\leq \lambda$. By doing so, we trace a score that is equal to the accumulative length of all seeds between both indices. For all non-overlapping SoCs, the index of the first seed (with respect to the SoC) in $A_S$ and the score (accumulative length of all seeds in SoC) are stored in a stack. This stack is finally transformed into a priority queue[44], where the order is defined by the SoC scores. This priority queue is then forwarded to the seed harmonization step. The pseudocode for the SoC computation can be found in Supplementary Note 6. The time complexity of the SoC computation is limited by the initial sorting. Figure 6 illustrates an exemplary SoC.

*Seed harmonization.* Seed harmonization computes a consistent subset of SoC seeds by applying a guided purging of contradicting seeds. Two seeds are consistent if they occur in the same order on query and reference. All seed in an alignment must be mutually consistent. For determining consistency, we introduce the concept of shadows for a given seed $s$: Informally, each seed is characterized by its start point and its end point in the two-dimensional plane spanned by query and reference (using the point based alignment representation introduced before). If we imagine a centered spotlight shining towards the seed, a shadow is created, where

the outer limits of the shadow are determined by the start and end-points defining the seed. A seed completely within the shadow of $s$ cannot be part of any consistent alignment comprising $s$. Note, that a partially shadowed seed is still consistent. Further, the spotlight can be placed on either side of $s$. This results in two shadows, one in its second and one in its fourth quadrant. We denote these shadows by $\sigma_{II}$ and $\sigma_{IV}$, respectively. The indexing scheme for shadows was chosen in accordance with the quadrant enumeration using Roman numerals in plane geometry. Figure 7 gives a visual example. Formally, we define:

$$\sigma_{II} := [0, r + l] \times [q, |Q|), \tag{2}$$

$$\sigma_{IV} := [r, |R|] \times [0, q + l). \tag{3}$$

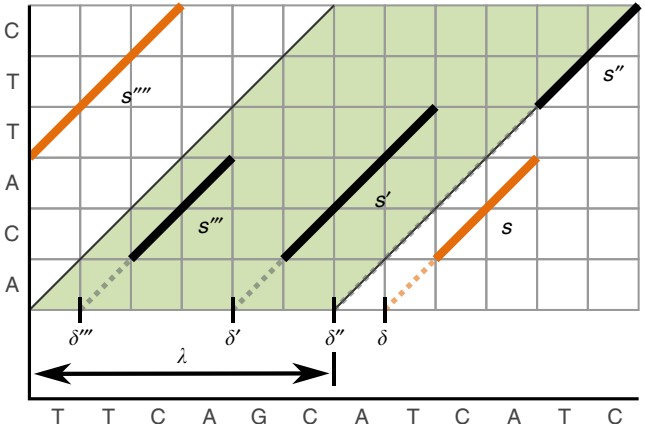

**Fig. 6** Strip of consideration (SoC). Seeds are visualized as solid lines. The SoC is the shaded area that comprises the seeds $s'$, $s''$ and $s'''$. The SoC width is equal to $\lambda$, where $\lambda$ is computed on the foundation of the query size and a given scoring scheme. The orange seeds $s$ and $s'''$ are outside of the SoC. The dashed line to the bottom-left of each seed leads to the respective $\delta$-value for this seed. The SoC score is the accumulated length of seeds within the SoC. In this example, the SoC score is seven

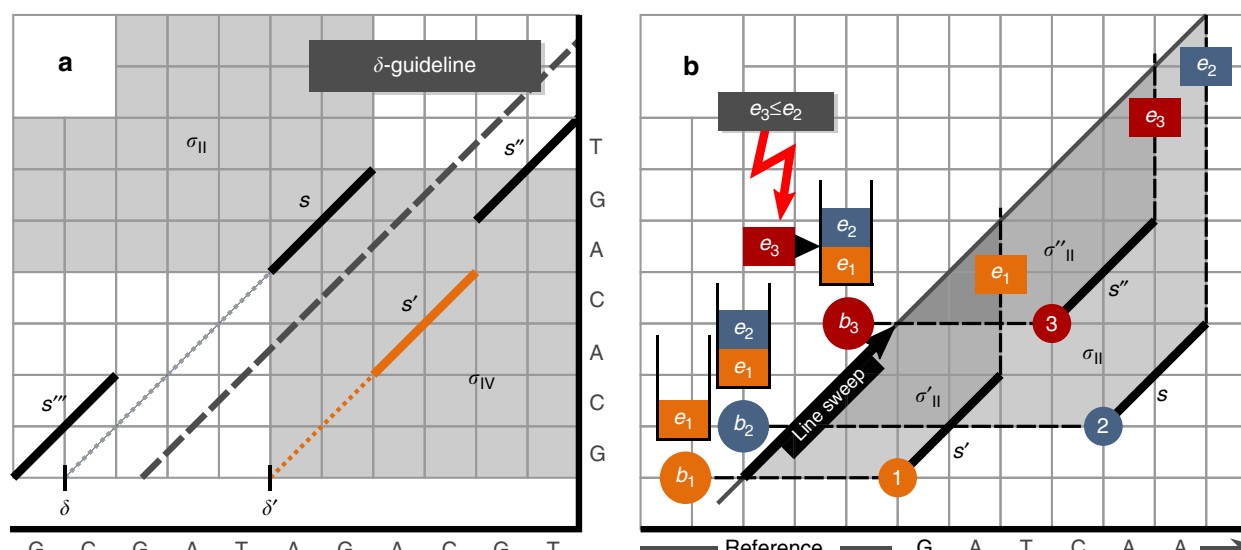

**Fig. 7** Seed-Shadows. **a** We consider a SoC with 4 seeds. The shadows spawned by the seed $s$ are displayed. The dashed line visualizes the guideline used in the context of the removal of conflicting seeds. $S_A = \{s, s'', s'''\}$ and $S_B = \{s', s'', s'''\}$ represent sets of mutually consistent seeds. $s$ and $s'$ shadow each other, so they are contradicting. $s'$ is deleted, because it is more distant to the $\delta$-guideline than $s$. So, $S_A$ is returned by the harmonization finally. **b** The figure shows three seeds and their corresponding shadows in the second quadrant. The shadow of $s$ fully encompasses the shadow of $s''$. So, these seeds are contradicting. If shadows are not overlapping or partially overlapping, the corresponding seeds are not contradicting. This is the case for $s'$, which is neither contradicting to $s$ nor $s''$. This shadow inclusion problem can be solved by using the line sweep principle as follows: Prior to the line sweep, we sort the seeds according to their start positions on the query. The line sweep visits the seeds in sorted order and stores their end positions on the reference via a stack. There is a contradiction if and only if the start positions on the query (these are given implicitly by the order of seeds) and the stored end positions on the reference are in different order. This is the case for $e_3$, which must occur after $e_2$, if there shall be no contradiction

Note, by using this definition $\sigma_{II}$ and $\sigma_{IV}$ are partially overlapping.

In the context of the seed harmonization, we compute a $\delta$-guideline on the foundation of all seeds in a SoC. The $\delta$-guideline forms a decision tool during the purging of conflicting seeds. For the computation of the $\delta$-guideline, we create a point cloud $P_{SoC}$ in the plane spanned by query and reference. $P_{SoC}$ comprises the start point, center point and end point of all seeds of a SoC. Using random sample consensus (RANSAC)[45], the angle and position of the $\delta$-guideline is computed on the foundation of the point cloud $P_{SoC}$. For a pair of inconsistent seeds, we measure their proximity to the $\delta$-guideline and delete the more distant one.

In detail, the $\delta$-guideline based removal happens according to the following algorithmic scheme: Let $\sigma_{II}$ ($\sigma_{IV}$) and $\sigma'_{II}$($\sigma'_{IV}$) be the second (fourth) quadrant shadows of two seeds $s = (q, r, l)$ and $s' = (q', r', l')$, respectively. Our approach is based on the observation that $s'$ is within $\sigma_{II}$ if and only if $\sigma'_{II}$ is encompassed by $\sigma_{II}$. Accordingly, $s'$ is within $\sigma_{IV}$ if and only if $\sigma'_{IV}$ is encompassed by $\sigma_{IV}$. $\sigma'_{II}$ within $\sigma_{II}$ is determined by (a) $q \leq q'$ and $r' + l' \leq r + l$; $\sigma'_{IV}$ within $\sigma_{IV}$ is determined by (b) $r \leq r'$ and $q' + l' \leq q + l$. The seeds $s$ and $s'$ are conflicting if and only if $\sigma'_{II}$ is within $\sigma_{II}$ or $\sigma'_{IV}$ is within $\sigma_{IV}$. As described above, we keep the seed closer to the $\delta$-guideline and remove the other.

(a) and (b) represent instances of the interval-interval-inclusion problem. We call our problem interval-interval-inclusion problem for avoiding confusion with the notion interval-inclusion problem. The latter one is usually used in the context of point inclusion decisions. However, we will do enclosure decisions with respect to complete intervals here. The enclosure of complete intervals can be efficiently solved using the line sweep paradigm. Using an example, Fig. 7 visually describes the line sweep-based discovery of contradicting seeds by means of their shadows. All details including pseudocode can be found in Supplementary Note 7.

The asymptotic worst-case complexity of the seed harmonization is $O(|S|^2)$, where $|S|$ is the number of seeds in a SoC. More detailed, the time complexity can be expressed as $O(|S| \cdot \xi + |S|)$ with $0 \leq \xi \leq |S|$, where $\xi$ is smaller than or equal to the number of contradictions of the seed with the most contradictions. Practically, however, we observed that the initial sorting of the harmonization is the driving runtime factor.

**Dynamic programming**. There can still be gaps between seeds after harmonization. In order to fill these gaps, we rely on a bandwidth-limited variant of dynamic programming (Needleman-Wunsch[25] with affine gap penalties). This represents a standard approach used by other aligners as well[7,9]. Further, the first seed might not touch the query start and the last seed might not extend to the query end. In this case, we rely on an adaptive banded semi global dynamic programming approach. Practically, we relied on ksw2 (the algorithmic approach was proposed by Gotoh[35]) for filling gaps and libGaba (based on the algorithmic approach detailed in Suzuki et al.[36]) for performing semi global extensions.

**Heuristic optimizations**. Our approach relies on several heuristic optimizations for getting an improved runtime behavior. (1) After seeding, we probe the periodic occurrence of sufficiently sized seeds and give up on failing queries. Our approach is accumulative here. Formally, we require $\sum_{s \in S} \frac{|s|}{\alpha} \geq \frac{|Q|}{\beta}$, where $\alpha$ is a minimum seed length, $\beta$ is a segment size, $Q$ the given query and $S$ the set of seeds for $Q$. For each segment, we expect one sufficiently sized seed in $S$ at least. Once a query passed this test, our approach delivers one alignment for this query at the minimum. (2) There are two user defined thresholds minSeedSize and maxAmbiguity. All seeds with a size below minSeedSize are removed. Further, if there are more than maxAmbiguity many seeds with identical query intervals, we purge them all. (3) During SoC computation, we purge SoCs using two user defined thresholds $\gamma$ and $\xi$. $\gamma$ represents an imaginary minimum seed length and $\xi$ ($0 < \xi \leq 1$) is a scalar factor with respect to the query length. For a query $Q$, we purge all SoCs with a score (accumulated seed length) below $\max(\gamma, \xi \cdot |Q|)$ (i.e., we demand seeds of minimal accumulative size $\gamma$ and $\xi \cdot |Q|$). Whenever we complete the harmonization of a SoC, we apply the above filter to the altered SoC. (4) The harmonization of consecutive SoCs happens according to the order given by the extraction from the priority queue. Depending on the query length, we apply two different heuristics, where the first one represents a filtering technique and the second one a stop-criteria. For long queries (roughly more than 1,000 nt), we always inspect a predetermined number of SoCs and discard SoCs that get a lower score than preceding SoCs. For shorter queries, we trace equality using a look-ahead approach. If we observe an unaltered score for more than a given threshold number of times, we break and move forward to DP. The latter two heuristics represent the outcome of an observational optimization. Improvements with respect to the filtering of harmonized SoC could be a valuable topic in the context of further research.

## Data availability

All data used in this study are available via the public sources listed below: GiaB HG002: ftp://ftp-trace.ncbi.nih.gov/giab/ftp/data/AshkenazimTrio/HG002_NA24385_son/. Homo sapiens GRCh38.p12 genome: https://www.ncbi.nlm.nih.gov/assembly. The underlying data of all diagrams can be created using our evaluation tool (https://github.com/ITBE-Lab/MA-EVAL) in combination with the above data sources.

## Code availability

MA—The Modular aligner—is available under the MIT license at https://github.com/ITBE-Lab/MA. The code for error profile sampling, read generation and aligner evaluation is available under the MIT license at https://github.com/ITBE-Lab/MA-EVAL.

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

## Acknowledgements

This research was supported by the Basic Science Research Program through the National Research Foundation of Korea (NRF) funded by the Ministry of Education (2016R1D1A1B03932599).

## Author contributions

The project was conceived by all authors. M.S. and A.K. devised and analyzed all algorithmic schemes with K.H. contributing valuable remarks. M.S. and A.K. implemented MA and conducted all experiments. The manuscript was written by M.S. and A.K. with assistance of K.H.

## Additional information

**Competing interests:** The authors declare no competing interests.

