## [Peer Review File · Nature Communications]

Reviewer #2 (Remarks to the Author):

The authors have addressed my comments from my previous review.

Reviewer #3 (Remarks to the Author):

General Comments:

We thank the authors for addressing our comments. With the extensive feedback from the other reviewers we feel that the manuscript is greatly improved. In particular, we find the analysis of aligner importance to SV alignment to be a very important addition (Especially the mapping consequences of split alignments vs precise alignment in repetitive regions). Due to the extensive changes, however, there are a few new issues that have arisen.

Major Comments:

- On page 5 the rationale used when explaining why mean values of error rather than positions is confusing and possibly wrong.
 - o Justification as to why one would simulate the error in a uniform fashion on long reads may be fairly pointless; it has been well documented that the error on PacBio sequencing is effectively random (not position or sequence-specific). Oxford Nanopore data is similar, with effectively random uniform error aside from some errors (e.g. homopolymers collapse, chimeric reads, low-quality head and tail regions). The “error profiles become highly variable towards the end because of the limited numbers of reads available for sampling” (Sedlazeck 2018, Supp. Note 3) and not because of “increasing noise towards the end” of the read as stated on page 5.
 - o Illumina data is more accurate near the beginning of the read and error will be concentrated near the end of the read as the cycles become more out of sync and the quality of the optical signal decreases. The paragraph implies that the authors simulated error uniformly which is not representative of true Illumina data.
 - o Although the method of simulation is much better than before, it seems strange that the authors did not simply use existing read simulation tools that are well tested and more closely mirror the error profiles in the datatypes but produce an effective ground truth. There are many published read simulators (e.g. PBSim, Nanosim, pIRS, ART, NEAT etc.) which can be tuned to different

scenarios, but the authors lack justification as to why they chose to develop their method for simulating reads and sufficient evidence that the simulated reads closely mimic read data.

- For example, the Sedlazeck et al chose to extend SURVIVOR and simulate their own long reads but had an accompanying evaluation of their simulation method to justify what they had performed. Note that they still used dwgsim for Illumina reads.

Minor Comments:

- Figure 2b makes almost no sense without Sup. Fig. 3 or note 3, if one is not familiar with the terms defined. If possible, a short an explanation of them should be included in the figure or main text.
- The abstract could use a rewrite. It has become more correct based on reviewer comments from the last revision but is more awkward to read. Try to include an emphasis on how it can be generically applied to a wide range of sequencing data, without compromising performance. MA's performance on aligning different types of structural variant indels should be mentioned somewhere in the abstract.
- We understand that some ideas/concepts may arise independently, but line-sweep algorithms have been used in other tools for seed chaining such as Mashmap2. It might be worth including some discussion or background on existing line-sweep algorithms or used in alignment, and explain how your approach relates to existing methods.

Response to referees (revision 2)

First, we would like to thank referee 2 and 3 for the work and time that they spent on our manuscript and the additive feedback.

Referee's 3 concerns – point-by-point response

Red italic text shows the reviewer's original response. Black text contains our notes addressing the reviewer's concerns.

Major issues

- On page 5 the rationale used when explaining why mean values of error rather than positions is confusing and possibly wrong. Justification as to why one would simulate the error in a uniform fashion on long reads may be fairly pointless; it has been well documented that the error on PacBio sequencing is effectively random (not position or sequence-specific). Oxford Nanopore data is similar, with effectively random uniform error aside from some errors (e.g. homopolymers collapse, chimeric reads, low-quality head and tail regions). The "error profiles become highly variable towards the end because of the limited numbers of reads available for sampling" (Sedlazeck 2018, Supp. Note 3) and not because of "increasing noise towards the end" of the read as stated on page 5.*

Fixed:

We agree with the reviewer. Therefore, we removed the confusing/wrong statements. More statements with respect to our read generation scheme are given in major issue 3 of this response letter.

- Illumina data is more accurate near the beginning of the read and error will be concentrated near the end of the read as the cycles become more out of sync and the quality of the optical signal decreases. The paragraph implies that the authors simulated error uniformly which is not representative of true Illumina data. [...] Note that they still used dwgsim for Illumina reads.*

Fixed:

We removed our original analysis of Illumina reads and inserted a new analysis on the foundation of DWGSIM (<https://github.com/nh13/DWGSIM>). By doing so, the focus moved towards an analysis that integrates the mapping quality. Fig. 3 was motivated by corresponding diagrams of Li¹. The horizontal mirroring of our diagram (compared to Li's diagrams) is due to our personal preference to have the open end of the log-scale on the right side of the x-axis. In our first revision, we already mentioned that BWA-MEM is slightly superior to MA for short reads and that Bowtie 2 is slightly inferior to MA. The new Fig. 3 confirms these statements.

Further, we added a discussion with respect to Illumina reads in Supp. Note 2.

- 3. Although the method of simulation is much better than before, it seems strange that the authors did not simply use existing read simulation tools that are well tested and more closely mirror the error profiles in the datatypes but produce an effective ground truth. There are many published read simulators (e.g. PBSim, Nanosim, pIRS, ART, NEAT etc.) which can be tuned to different scenarios, but the authors lack justification as to why they chose to develop their method for simulating reads and sufficient evidence that the simulated reads closely mimic read data.*

- For example, the Sedlazeck et al chose to extend SURVIVOR and simulate their own long reads but had an accompanying evaluation of their simulation method to justify what they had performed. Note that they still used dwgsim for Illumina reads.*

Answer:

First, we would like to explain the motivation for an analysis on the foundation of our own simulation scheme:

In order to realize the manuscript's Fig. 2, it is necessary to have a read simulator that can create sets of reads with specific (fix) mean error rates (for insertions, deletions and substitutions). By scaling these error rates, it becomes possible to visualize an aligner's mapping characteristic with respect to reads of different qualities.

To our knowledge, there is no generator for PacBio reads / UON reads that offers this opportunity as build-in-feature. Instead, they try to generate reads that span over the complete spectrum of error rates (low quality reads as well as high quality reads together in one set) without any possibility to assign the reads to specific buckets representing specific error rates.

In order to assess our simulation approach, we performed a principal component analysis (PCA) for our read generation scheme as part of Supp. Note 2.

Additional to the analysis over there, we now directly assess the simulators SURVIVOR and NanoSim in the context of original reads (PacBio and UON).

Principal Component Analysis for SURVIVOR on PacBio reads

The right plot below shows a PCA for the generator SURVIVOR.

The left plot of the above figure is identical to Supp. Fig. 2 a.

The SURVIVOR reads occur on a narrow band that lies within the lower triangle. All other areas of Supp. Fig. 2 a (this plot represents the original PacBio reads) are empty in the plot for SURVIVOR reads. Therefore, SURVIVOR creates PacBio-like reads of various lengths but only within a limited band, omitting many others. MA-Eval represents the spectrum of original reads much more accurately as indicated by Supp. Fig. 2 d. Further, MA-Eval allows a ‘travel’ throughout the spectrum. By moving within the spectrum, the mapping of reads becomes more and less challenging for aligners, which is in turn reflected by different amounts of accuracy that are measured in Fig. 2 of the manuscript.

In order to show our observations from another perspective, we insert the IGV screenshot below. The screenshot compares original PacBio reads of some random area of the human genome (top) with simulated reads of our generation scheme (MA-Eval in the center) and reads generated by SURVIVOR (bottom):

IGV screenshots of generated reads (PacBio original, our scheme, SURVIVOR)

The IGV settings in the above screenshot suppress the display of all indels with length <10nt. The top plot clearly indicates the existence of indels ≥ 10 nt in original PacBio reads. In the bottom plot for SURVIVOR, there appears one location merely with an indel of size ≥ 10 nt. In contrast, the plot for MA-Eval (center) shows a similar amount of occurrences as in the plot for original PacBio reads (top plot). This confirms the insights resulting from the PCA.

Principal Component Analysis for NanoSim on Ultralong Oxford Nanopore reads

The left plot of the above figure is identical to Supp. Fig. 3 a.

The PCA for NanoSim reads shows a similar picture as the PCA for SURVIVOR with PacBio reads. Once again, the simulated reads are merely within a small band, and the top triangle is completely unpopulated. MA-Eval represents the spectrum of original reads much more accurately as indicated by Supp. Fig. 3 d.

Minor issues

4. *Figure 2b makes almost no sense without Sup. Fig. 3 or note 3, if one is not familiar with the terms defined. If possible, a short an explanation of them should be included in the figure or main text.*

Fixed:

We agree with the reviewer. Fig. 2b of the first revision's manuscript and Supp. Fig. 3 are merged now. Together, they form Fig. 4 of the revised manuscript. A brief textual description of the different categories of alignments (precise, split etc.) remains in the manuscript. The detailed description stays in Supp. Note 3.

5. *The abstract could use a rewrite. It has become more correct based on reviewer comments from the last revision but is more awkward to read. Try to include an emphasis on how it can be generically applied to a wide range of sequencing data, without compromising performance. MA's performance on aligning different types of structural variant indels should be mentioned somewhere in the abstract.*

Fixed:

We agree with the reviewer and rewrote parts of the abstract. The abstract now consists of three segments: 1) Introducing sentence 2) Features and performance of proposed approach 3) Algorithmic highlights.

6. *We understand that some ideas/concepts may arise independently, but line-sweep algorithms have been used in other tools for seed chaining such as Mashmap2. It might be worth including some discussion or background on existing line-sweep algorithms or used in alignment, and explain how your approach relates to existing methods.*

Answer:

Mashmap2 relies on the line-sweep paradigm in the context of seed processing as MA does. Apart from notable algorithmic differences (e.g. usage of a binary search tree (Mashmap2) instead of a stack (MA)), the seed processing of Mashmap 2 is not designed for the recognition of conflicting seeds as it is needed by MA. (By our definition: Two seeds are conflicting, if they cannot be part of the same alignment). However, we inserted an additional note with respect to Mashmap 2 and seed processing into the introduction of the manuscript.

References

- 1 Li, H. Aligning sequence reads, clone sequences and assembly contigs with BWA-MEM. *arXiv preprint arXiv:1303.3997* (2013).

Reviewer #3 (Remarks to the Author):

All our comments are properly addressed.

The authors went above and beyond regarding the PCA analysis of their long reads. The MAPQ information is interesting, showing some room for improvement of calculation/calibration of the MAPQ (future work).

Response to the referees (final revision)

Red italic text shows the reviewer's original response. Black text contains our notes addressing the reviewer's concerns.

Reviewer #3 (Remarks to the Author):

All our comments are properly addressed.

The authors went above and beyond regarding the PCA analysis of their long reads. The MAPQ information is interesting, showing some room for improvement of calculation/calibration of the MAPQ (future work).

Answer:

Once more, we would like to thank reviewer 3 for the time and efforts invested into reviewing our submission!